# The effect of nursing shared governance educational program on nurse managers' knowledge for sustainable nursing excellence and empowerment

Sally Mohammed Farghaly Abdelaliem●¹*, Khalid Al-Mugheed², Majdi M. Alzoubi³, Islam Al-Oweidat⁴, Ghada Mohamed Hamouda⁵

1 Department of Nursing Management and Education, College of Nursing, Princess Nourah bint Abdulrahman University, Riyadh, Saudi Arabia, 2 Health Faculty, Nursing Department, Riyadh Elm University, Riyadh, Saudi Arabia, 3 Faculty of Nursing, Alzaytoonah University of Jordan, Amman, Jordan, 4 Nursing Administration, Faculty of Nursing, Department of Community and Mental Health Nursing, Head of Department, Zarqa University, Jordan, Jordan, 5 Public Health Nursing Department/Nursing Administration, Faculty of Nursing, King Abdulaziz University, Jeddah, Saudi Arabia

* Sally.farghaly@alexu.edu.eg, smfarghaly@pnu.edu.sa

## Abstract

### Background

Nursing governance residues one of the most composite functions in the complex and complicated healthcare arena. Providing nurse managers with professional nursing governance knowledge and its related practices is essential in today's competitive healthcare environment for nursing practices excellence.

### Objective

This research aimed to determine the effect of Professional Nursing Shared Governance Awareness Sessions on First-Line Nurse Managers' Knowledge.

### Design

A quasi-experimental research design.

### Setting

Alexandria Medical Research Institute hospital.

### Participants

A total of 50 first-line nurse managers were recruited for the participating hospital.

### Methods

Professional Nursing Shared Governance Knowledge Questionnaire and the participants' socio-demographic characteristics questions were included.

**Data availability statement:** All relevant data are within the manuscript.

**Funding:** The research was funded by Princess Nourah bint Abdulrahman University Researchers Supporting Project number (PNURSP2025R844), Princess Nourah bint Abdulrahman University, Riyadh, Saudi Arabia. The recipient of the funding Award is the author: Sally Mohammed Farghaly Abdelaliem. The funders had no role in study design, data collection and analysis, decision to publish, or preparation of the manuscript.

**Competing interests:** The authors have declared that no competing interests exist.

## Results

There was a statistically significant differences between the overall mean score of first-line nurse managers' knowledge about professional nursing shared governance before and after attending nursing shared governance awareness sessions (p < 0.001). The overall mean score of professional nursing shared governance was increased significantly from $3.12 \pm 2.54$ to $5.86 \pm 0.35$ (t 7.493, p < 0.001).

## Conclusions

This research plugs the Arabian nursing literature gap by representing how effective professional nursing shared governance awareness sessions could improve the nurse managers' level of knowledge regarding the governance concept and its practices. Participants perceived the awareness sessions as constructive and strategic contrivances to determine and evolve high-potential personnel in opportunities as giving compassionate organizational structures and effective leadership behaviors for aggregating contribution of nurses in work enterprise, problem-solving, conflict resolution, teams and organizational decision-making as "crucial components to an efficacious organization". Hence, the definitive issue for the nursing profession future is participative decision making through shared governance, as it should be a part of every organization's strategic plan.

## 1. Introduction

In complex health care environments, there are many obstacles to overcome, such as a lack of nurses, a shrinking labor force, a rise in patient acuity, and higher regulations that increase the workload of nurses, make them unhappy at work, and cut down on the amount of time they spend at the bedside. These issues increase the responsibilities and accountability of nurses but do not give them more authority or control to specify the necessary adjustments that will have an impact on nursing practice [1–4]. By implementing professional nursing governance and shared governance models and creating a beautiful workplace, healthcare managers and leaders realized that quality care is best delivered by nurses who are committed to their workplace and allowed by their leaders to exercise their profession with no restrictions and complete independence.

### 1.1. Theoretical framework

Professional nursing governance is a multifaceted concept that encompasses the structure and procedure that professional nurses use to direct, govern, and organize the various goal-oriented practices of their professional performance. It also affects the organizational perspective in which it occurs through organizational recognition, facilitation structures, the connection of data, and the configuration of objectives [5]. According to Francis-Johnson et al. (2018) [6], professional nursing shared governance entails giving frontline nurses more authority, power, and control over their operational performance.

A system that allows the nurse's viewpoint to be taken into account is known as "shared governance" and refers to a model of professional nursing practice that incorporates nurses and their supervisors in decisions that influence their performance [7]. It is a fundamental component of the clinical governance program and is necessary for good leadership and the creation of a learning environment at work. According to numerous studies [8–9], it is advantageous to authorize practitioners from a variety of contexts because it provides a structure for

healthcare professionals to work as a team and to foster multi-professional care. The organization's people needs may be met by modifying the structure, which could also assist the development of professional autonomy.

According to Burgess, M. M. (2014) [10], the shared governance ideology is linked to decentralized management, which fosters an empowering work environment. Partnership, equity, responsibility, and ownership are the guiding concepts of shared governance that were first foreseen by Dr. Tim Porter O'Grady [11–13]. In order to foster relationships between nurses and the multidisciplinary team, partnership is precarious. Equity shows that each team member is essential to providing patients with high-quality care. Accountability is the cornerstone of shared governance, and it calls for nurses to take risks when making decisions. According to Hess and Swihart (2013) [14], ownership involves adapting the professional job, where it is done, and who does it. Nursing leaders and hospital administration must encourage the nurses' engagement in decision-making if nursing shared governance is to be effective in a healthcare organization. By "connecting or affecting a judgment or decision" and "the pattern of distributing authority for decisions and actions that control nursing practice strategies and the practical workplace," respectively, Liu, Hsu, & Chen (2015) [15] defined decisional participation.

Distribution of authority, autonomy, empowerment, cooperation, responsibility, and accountability are the six descriptors of decisional involvement [15]. This model of decisional involvement incorporates Porter O'Grady's original four shared governance concepts: partnership, equity, accountability, and ownership [11]. The antecedents to decisional participation are also described by Kowalik and Yoder. These preconditions are fundamental events that must take place in order to obtain the desired conclusion. The shared governance council or committee's structure, the choice of nurses to participate in decision-making, and nurse authority over practice are the forerunners. Committees are created in healthcare settings to encourage and support nurse participation, but nurses must choose to participate and take responsibility for their participation. Enhanced nurse satisfaction, recruitment, and retention are further advantages of shared governance, as are decreased nurse absenteeism and turnover [16].

By shifting away from the traditional hierarchical management style to one where nurses are more involved in decision-making practices and managers have a facilitative, rather than dominating, role. Geoghegan and Farrington (1995) [17] discussed the benefits of this tactic, identifying that it contributes to health care professionals' collective responsibility and accountability for practice. They stress that doing so will increase self-assurance, job satisfaction, motivation, nurses' influence, creativity, support for interpersonal connections, ownership, and provide a sense of worth. Additionally, shared governance may promote collaborative relationships, improve clinical effectiveness and quality of care, boost nurses' self-confidence, support personal and professional development, raise their profile, encourage data sharing, promote effective communication, allow for the development of innovative skills and knowledge, increase professionalism and accountability, improve focus and direction, and reduce work duplication [16–17]. This may have a major impact on hiring and retaining new employees as well as improving professional and individual performance.

The development of effective shared governance takes time and thought. Shared governance is a way of life, not a project with a set deadline. Since each nursing unit and department should find the structure and procedure that works best, there is no "one size fits all" model. In a similar vein, Porter-O'Grady (2012) [11] notes that implementing an empowered format, such as shared governance, made clear that relationships, conclusions, structures, and procedures will constantly change at every level of the system. According to Ballard et al. (2014) [18], the cornerstones of effective shared governance are leadership provision, role

definition, decision-making procedures, a clear vision, communication tactics, education, and time to contribute.

For shared governance to be successful, Golanowski, Beaudry, Kurz, Laffey, and Hook (2007) [19] identified two key characteristics. The first is that decisions must be made at the point of care, and the second is that the organization must be clearly structured from the perspective of care so that all systems and processes support patient care. Failure of shared governance may occur from any of the following [18]: a lack of communication, a lack of support, a lack of education, or a lack of resources. A shared governance system's success or failure heavily depends on nurse managers. This is viewed as a significant challenge. Professional nursing shared governance hence cannot be put into effect right away. It demands a significant amount of commitment, leadership awareness of this idea and how to change their management style in conjunction with their nurses, as well as a great deal of agreement of attention and efficient preparation. It should never be put into effect as an immediate remedy.

The cumulative effects of health care demand new leadership strategies that achieve organizational objectives while creating and promoting healthy work environments. Additionally, empowering nurses to participate in decision-making with their nurse manager frequently depends on insufficient training, on-the-job experience, or limited access to the necessary information and resources with limited power, posing a risk to patient safety, satisfaction, and outcomes. A nurse manager who is ineffective will also threaten nurse confidence, retention, and turnover. First-line nurse managers has to complete a professional nursing shared governance process with an eye toward the future to ensure opportunity permutation and velvet protocols both inside the organization and throughout the nursing domain [20]. To alert nurse leaders to problems and challenges with the current structure and provide them with priority-oriented areas of improvement for the pursuit of excellence in nursing practices, it is necessary to evaluate nurse managers' knowledge of professional nursing shared governance. The evaluation's study findings can be utilized to motivate nurse managers to implement and promote a plan to rehabilitate and include nurses in nursing shared governance practices throughout the hospital.

Additionally, the initial training in professional nursing shared governance for nurse managers will help identify any knowledge gaps and assess their current management techniques. This is a timely area of research that can give leaders the knowledge they need to improve the current state of nursing professional governance and prioritize the necessary initiatives to improve the quality of care and advance the nursing profession. Formal nurse manager programs of professional nursing shared governance have been shown to increase nurse manager retention and capability rates, as well as helping to identify and develop new generations of nurse leaders, by successfully emerging and utilizing their current manpower resources [5]. This affirms the requirement for the development and maintenance of nurse managers through shared governance initiatives and practices in the field of professional nursing.

Numerous studies that examined nurses' perceptions of shared governance were published in West in the preceding years. Previous studies confirmed the variations in nurses' perceptions of and participation in their professional nursing practice and governance and recommended additional investigation into and evaluation of professional nursing governance in broad and varied situations [21–23]. Although professional nursing governance has received a lot of attention, there hasn't been any published research that clearly assesses the level of understanding of the professional nursing shared governance idea among nurse leaders. Precisely to what extent the professional nursing shared governance baseline knowledge among nurse managers is presently unidentified. It is essential to enrich the present knowledge and practices about professional nursing shared governance at hospitals to update nurse leaders of concerns and issues within the current hospital and nursing structure and afford them priority-oriented

areas of advance for the reaching to nursing excellence. Additionally, the nurse leaders of the study setting expressed their interest to take a training on the professional shared governance while conducting the previous research study on developing a strategy for professional nursing governance after the baseline assessment and gap analysis and developing the strategy based on their level of applying the practice of shared governance. This research revealed that the gap analysis results detected (145 strength points and 79 weak points need for improvement). Also, it was found that nurses apply the first level of nursing shared governance. While, the perception of professional nursing governance was traditional management as perceived by nurses [24]. Therefore, the purpose of this research is to determine the effect of Professional Nursing Shared Governance Awareness Sessions on First-Line Nurse Managers' Knowledge. The PICO research question addressed by the current study was:

**P**: First-Line nurse managers at Alexandria Medical Research Institute hospital.

**I**: Professional Nursing Shared Governance Awareness Sessions

**C**: No comparator

**O**: Increased first-line nurse managers' knowledge about professional nursing shared governance concept as measured by the developed Professional Nursing Shared Governance Questionnaire.

The **PICO** question is: Does a Professional Nursing Shared Governance Awareness Sessions increase the first-line nurse managers' knowledge about professional nursing shared governance concept, at Alexandria Medical Research Institute hospital?

The working hypotheses were the following:

($H_1$) First-line nurse managers' knowledge will be significantly increased by their participation in Professional Nursing Shared Governance Awareness Sessions.

($H_2$) First-line nurse managers' knowledge will be the same after their participation in Professional Nursing Shared Governance Awareness Sessions.

## 2. Methods

### 2.1. Research design and Setting

The researchers accompanied a quasi-experimental research design at Alexandria Medical Research Institute hospital with a bed capacity of 568 beds at Egypt which is affiliated with the university health sector and identified as teaching hospital that provide a wide range of medical care (primary, secondary and tertiary care), education, training, and research services and it applies the governance standards through the health governance unit.

### 2.2. Subjects

The study's participants were chosen using a whole population sampling strategy (purposive sampling technique), and they included all first-line nurse managers who were employed by the hospital and available during the data collection period (N = 50 out of 54), with the aim of assessing their understanding of the concept of professional nursing shared governance and their readiness to take part in the study. All first-line nurse managers working in the study setting met the inclusion criteria, while any staff nurse without a management nursing position did not.

### 2.3. Measurements

#### 2.3.1. Professional nursing shared governance knowledge questionnaire. The researchers developed this questionnaire constructed on reviewing the associated literature intended for assessing first-line nurse managers' knowledge of professional nursing shared

governance concept before and after awareness sessions [10–12,20–24]. It comprises 10 several response formats questions. Responses to question 1,9,10 by Yes (1) as well as No (0); starting from questions 3 to 8 responses evaluated on 4-point Likert scale ranging from completely know (4) to do not know (1). A higher score showed virtuous knowledge. It asks about the knowledge of definitions, purposes, principles, elements, strategies, models, techniques, guidelines, characteristics, benefits and challenges of professional nursing shared governance. The Cronbach's alpha value of the questionnaire was 0.89 which was reliable.

**2.3.2. Evaluation of participants' satisfaction about professional nursing shared governance awareness sessions.** The participants' satisfaction was evaluated using an evaluation form comprised 10 items based on the related literature [10–12]. Responses weighted on a Likert scale with five-points extending from strongly agree (5) to strongly disagree (1). A higher score meaning a higher level of satisfaction.

Besides, the researchers developed demographic characteristics form to evolve data on first-line nurse managers' socio-demographic and working associated characteristics. It encompassed questions regarding; gender, age in years, working unit, educational level, and years of experience.

## 2.4. Validity and reliability

**2.4.1. Tool adaptation.** The study's instruments were created in English and then translated into Arabic using a forward-backward translation technique (Alyami et al. 2019). Five academic experts, whose areas of expertise included nursing administration and health governance, reviewed the tools for translation, face and content validity, and relevance. As a result, the questionnaire underwent little changes. To assess their validity and reliability, we used a variety of techniques. For example, we developed the coding instrument by applying Steps 1–6 of DeVellis' (2017) paradigm for scale construction (refer to Fig 1). Reviewing the literature study [10–12,20–24] helped to clarify the construct and establish an item pool. Furthermore, Cronbach's alpha and content validity were used.

**2.4.2. Validity of content.** Five experts from the field, comprising two professors of nursing administration and three professors of health governance. The panel found typographical, punctuation, and word choice issues. In response to their recommendations, a few terms were modified. The knowledge questionnaire's stability over time was investigated in a pilot study with ten nurse supervisors to confirm the instruments' accuracy and functionality and to ascertain the time required to complete the questionnaires. The results revealed a high positive significant correlation (r ranged from 0.782 to 0.890). The pilot sample was absent from the study sample.

**2.4.3. Internal consistency.** In order to evaluate the instruments' internal consistency, Cronbach's alpha was employed. It is a gauge for how well a collection of items represents a particular concept or dimension. more numbers indicate more reliability and internal consistency. The range is 0 to 1. The professional nursing shared governance knowledge questionnaire had a Cronbach's alpha rating of 0.89. This outcome shows that the study instruments' internal consistency and dependability are satisfactory.

## 2.5. Methods

The Pre-Awareness and Post-Awareness Phases were the two primary stages of the study's execution. Pre-test, content development, and Professional Nursing Shared Governance Awareness Sessions were all part of the pre-awareness phase. Post-awareness phase activities included data analysis and post-test. Before the execution of the awareness sessions, there was a pre-test consuming knowledge questionnaire for evaluating first-line nurse managers'

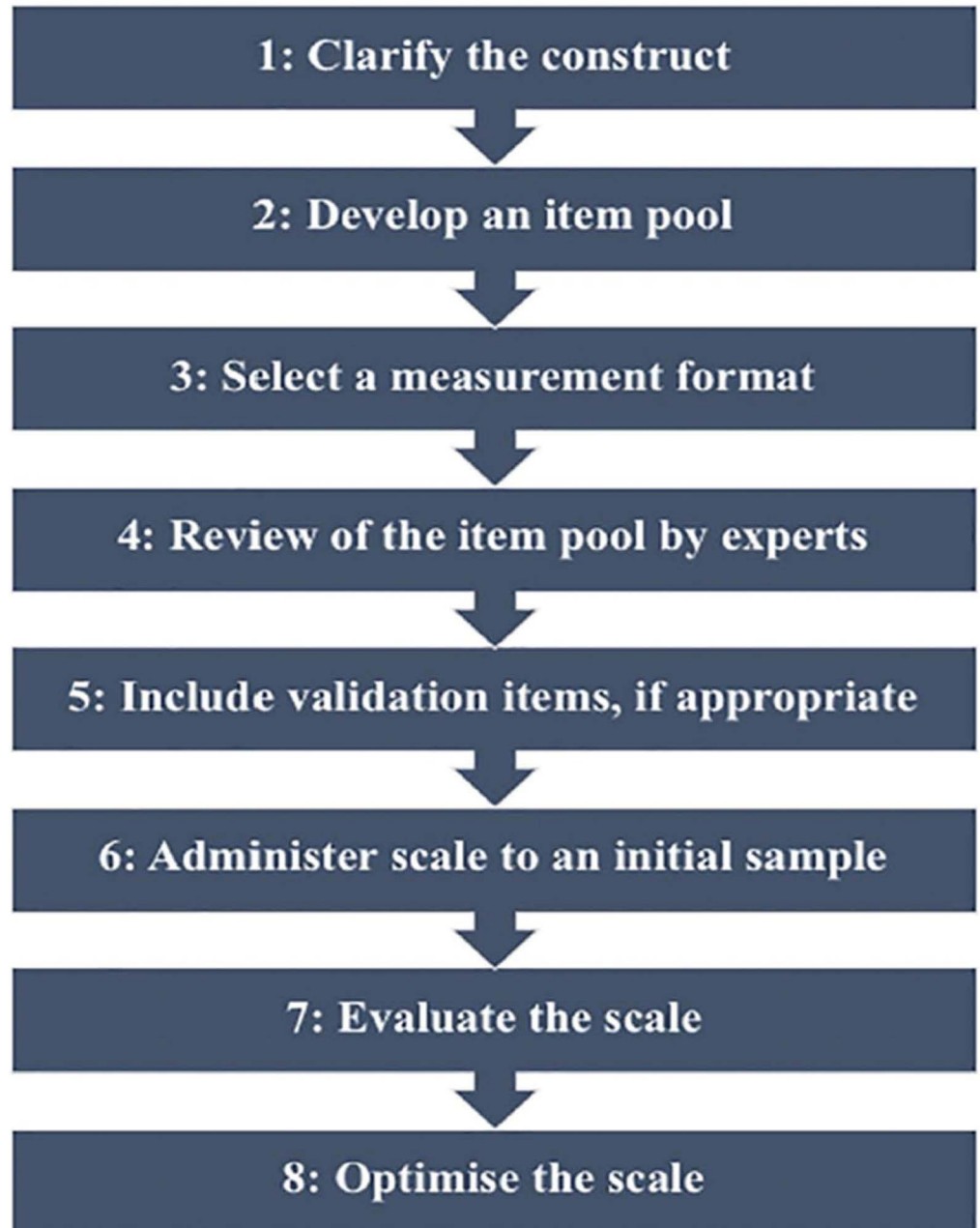

**Fig 1. Steps in the Scale Development (DeVellis, 2017; El Demerdash et al., 2024).**

knowledge of the professional nursing shared governance concept. The content of the awareness sessions was developed based on a review of the relevant literature and the learning needs of the managers as determined by the results of the pre-test. The same specialists that validated the study methods also assessed a handout on awareness. The final handout was then created and presented throughout the awareness sessions after a few minor changes. In order to avoid interfering with their work schedules and the coordination of patient care, first-line nurse supervisors decided on the time and location for the sessions while taking into account their duty scheduling and vacation days.

The educational workshops were given to increase their understanding of the notion of professional nursing shared governance. The nurse managers were divided into 5 groups, each of which included 10 Managers, because it was not possible to hold the awareness sessions with all of them at the same time as had been arranged. Three consecutive awareness sessions (weekly sessions) were given to each group in order to cover the topics covered in the sessions. Each treatment took around two hours.

The researchers established a connection with front-line nurse managers during the first session, when they discussed professional nursing shared governance, its definition, general overview, principles, elements, and advantages. In the second session, the researchers educated nurse managers on professional nursing shared governance models and methods and discussed the benefits and drawbacks of implementing it in their hospitals. In the third session, participants provided feedback by reflecting on the previous workshops while also being introduced to common practices, guidelines, and characteristics of adopting professional nursing shared governance.

With respect to the information offered, the content was gathered using appropriate and cooperative teaching techniques, including (lecture, small group discussion, video presentation, and brainstorming). To increase their understanding, first-line nurse supervisors received the awareness pamphlet. At the hospital where the study was conducted, around 15 sessions were held to cover all the material. Sessions on shared governance for the profession of nursing were conducted in the conference room. After finishing their job, the majority of the sessions were implemented during morning duty, while a small number were implemented during evening duty. The third session of the post-awareness phase's post-test, which used a shared governance knowledge questionnaire to assess first-line nurse managers' knowledge of the concept of professional nursing shared governance and gauge their satisfaction with the awareness sessions, was immediately introduced for each group. In the end, data analysis was done. (See Data analysis section) (Table 1).

## 2.6. Data collection

The administrator of the recognized hospital gave the written consent for data collection. The researchers conducted the awareness sessions and data collection using the created materials and questionnaires distributed individually to nurse managers before to and

**Table 1. Description of the intervention.**

| Phases | Actions | Tools/ methods Used |
|---|---|---|
| Pre-awareness phase | • Pre-test | Professional nursing shared governance knowledge questionnaire. |
| | • Awareness sessions Content Development | Nurse Managers' learning requirements, and related literature Content reviewed by experts in the study field. |
| | • Intervention implementation | **Content:** <br>• Broad and specific objectives of nursing shared governance. <br>• Professional nursing shared governance definition and general review. <br>• Professional nursing shared governance purposes, and principles <br>• Elements of Professional nursing shared governance. <br>• Strategies, models and techniques of nursing shared governance. <br>• Guidelines of Professional nursing shared governance. <br>• Characteristics of Professional nursing shared governance. <br>• Benefits and challenges of Professional nursing shared governance. <br>• Reflective practice. |
| Post-awareness phase | • Post-test | • Professional nursing shared governance knowledge questionnaire. |
| | • Data Analysis | Compare pre-test with post-test results for knowledge questionnaire. |

during the awareness sessions. The study questionnaire took each participant 10 to 20 minutes to complete. With the support of nurse managers, data collection and awareness training about shredding governance were coordinated to take place at the same time. From June to August 2022, three months were spent on data gathering and delivery of awareness interventions.

### 2.7. Ethical considerations

The aim of the study was explained to subjects and written informed consent was obtained from them for their participation in the study and from the Princess Nourah bint Abdulrahman University—Institutional Review Board (IRB) approval (H-01-R-059). Confidentiality, privacy and anonymity of subjects were sustained and guaranteed. The study subjects were guaranteed they ensured the right to take away from the study. The inclusion and exclusion criteria prevented bias in sample selection, and the entire population was already taken into account by withholding the data of participants who declined to participate and showing respect for their opinions by withholding their own data.

### 2.8. Data analysis

Data were coded and nourished to the statistical package of social science (IBM SPSS), version 22 and IBM AMOS version 23. For presenting socio-demographic characteristics, frequencies, and percentages were utilized. Mean and standard deviation (SD) were utilized to existing continuous variables. The effect of awareness sessions was examined with the parametric test (paired t-test) for normally distributed data. As well as non-parametric tests as MH: Marginal Homogeneity test and McN: McNemar test were used for abnormally distributed data to compare the pre-test/post-test scores. The McNemar test is the best test for dichotomous variables with two dependent sample studies. When a category of the sample is more than two, marginal homogeneity tests are appropriate; they are essentially an extension of the McNemar test for dependent samples. Pearson correlation coefficient analysis (r) were utilized to test the relationship among variables.. All statistical analyses were accomplished using an alpha of 0.05.

## 3. Results

As regards the socio-demographics and work-related characteristics. All the studied first-line nurse managers were female. Slightly 44% of participants' age were ranged from 30 to less than 40 years. They were working in different units including Medical (20%), Surgical (30%), Infection Control (12%), ICUs (24%), Quality Management (24%), OR/Recovery (16%), Oncology (12%), and Nursing Management (12%). More than two thirds of them (62%) held a Bachelor of Nursing Science. 47% of first-line nurse managers had nursing' years of experience ranged from ten to less than twenty (Table 2).

Table 3 shows first-line nurse managers' knowledge about professional nursing shared governance before and after shared governance awareness sessions. Most of participants (92%) reported that they had no previous information about professional nursing shared governance before awareness sessions, while all of them reported that they gained information about nursing shared governance after awareness sessions with a highly significant difference ([McN]p < 0.001). Significant improvements were also detected regarding first-line nurse manager responses for the professional nursing shared governance definition, purposes, principles, elements, benefits, challenges, strategies, techniques, and models, of professional nursing shared governance ([MH]p < 0.001). Most of first-line nurse managers

**Table 2. First-line nurse managers' distribution according to their demographic and work related characteristics (N = 50).**

| Socio-demographic and work related characteristics | No. | % |
|---|---|---|
| **Gender** | | |
| Male | 0 | 0.0 |
| Female | 50 | **100.0** |
| **Age (years)** | | |
| <30 | 6 | 12.0 |
| 30–<40 | 22 | **44.0** |
| 40–<50 | 19 | 38.0 |
| >50 | 3 | 6.0 |
| **Working Unit** | | |
| Medical | 10 | **20.0** |
| Surgical | 15 | **30.0** |
| Infection control | 3 | **12.0** |
| ICU | 6 | **24.0** |
| Quality Management | 6 | **24.0** |
| OR/Recovery | 4 | **16.0** |
| Oncology | 3 | **12.0** |
| Nursing Management Office | 3 | **12.0** |
| **Educational level** | | |
| Master Degree in Nursing | 9 | 18.0 |
| Bachelor of Nursing Science | 31 | **62.0** |
| Diploma of Technical Health Institute | 10 | 20.0 |
| Diploma of Secondary Technical Nursing | 0 | 0.0 |
| **Nursing years of experience** | | |
| 5–10 | 6 | 12.0 |
| 10–<20 | 25 | **47.0** |
| >20 | 19 | 41.0 |

(86%) confirmed that they need professional nursing shared governance training before and after attending the awareness sessions. All participants perceived the relation between professional nursing shared governance and nursing practices excellence. Concerning the overall evaluation of awareness sessions, the mean percent score 78.8% revealed participants' satisfaction with these sessions (3.94 ± 0.67).

Table 4 displays statistically significant differences between the overall mean score of professional nursing shared governance before and after awareness sessions (p < 0.001). The overall mean score of professional nursing shared governance was increased significantly from 3.12 ± 2.54 to 5.86 ± 0.35 (t 7.493, p < 0.001).

Concerning the overall evaluation of professional nursing shared governance awareness session, they were satisfied by the professional nursing shared governance awareness sessions with a mean score of 3.94 ± 0.67. Most of the first-line nurse managers were satisfied with the way the instructor taught in the sessions, the effective teaching and training methods used and the information they gained. They granted that sessions increased their strengths and self-awareness, increased their feeling of their leadership role toward their nurses, and recommended to repeat this session to all nurses to prepare them for professional and leadership roles and they agree that the professional nursing shared governance concept can be applied into nursing practice (Table 5).

**Table 3. First-line nurse managers' knowledge of professional nursing shared governance before and after awareness sessions (N = 50).**

| Awareness of Professional Nursing Shared Governance Variables and Related Dimensions | Before | | After | | p |
|---|---|---|---|---|---|
| | No. | % | No. | % | |
| **Previous information** | | | | | McNp < 0.001 * |
| Yes | 4 | 8.0 | 50 | 100 | |
| No | **46** | **92.0** | 0 | 0.0 | |
| **Sources of information** | | | | | McNp = 1.000 |
| From my Academic study | 0 | 0.0 | 0 | 0.0 | |
| Other sources (Webinar, and online courses) | 8 | 16.0 | 8 | 16.0 | |
| **Definition of professional nursing shared governance** | | | | | MHp < 0.001 * |
| Complete definition | 25 | 50.0 | 50 | 100 | |
| Incomplete definition | 2 | 4.0 | 0 | 0.0 | |
| Wrong answer | 7 | 14.0 | 0 | 0.0 | |
| Don't know | 16 | 32.0 | 0 | 0.0 | |
| **Purposes of Professional nursing shared governance** | | | | | MHp < 0.001 * |
| Correct/Complete answer | 25 | 50.0 | 50 | 100 | |
| Incorrect/Incomplete answer | 9 | 18.0 | 0 | 0.0 | |
| Wrong answer | 10 | 20.0 | 0 | 0.0 | |
| Don't know | 6 | 12.0 | 0 | 0.0 | |
| **Principles of Professional nursing shared governance** | | | | | MHp < 0.001* |
| Correct/Complete answer | 34 | 68.0 | 50 | 100 | |
| Incorrect/Incomplete answer | 7 | 14.0 | 0 | 0.0 | |
| Wrong answer | 4 | 8.0 | 0 | 0.0 | |
| Don't know | 5 | 10.0 | 0 | 0.0 | |
| **Elements of Professional nursing shared governance** | | | | | MHp < 0.001* |
| Correct/Complete answer | 28 | 56.0 | 50 | 100 | |
| Incorrect/Incomplete answer | 4 | 8.0 | 0 | 0.0 | |
| Wrong answer | 10 | 20.0 | 0 | 0.0 | |
| Don't know | 8 | 16.0 | 0 | 0.0 | |
| **Benefits and challenges of Professional nursing shared governance** | | | | | MHp < 0.001* |
| Correct/Complete answer | 28 | 56.0 | 47 | 94.0 | |
| Incorrect/Incomplete answer | 5 | 10.0 | 3 | 6.0 | |
| Wrong answer | 9 | 18.0 | 0 | 0.0 | |
| Don't know | 8 | 16.0 | 0 | 0.0 | |
| **Strategies/techniques/models of Professional nursing shared governance** | | | | | MHp < 0.001* |
| Correct/Complete answer | 16 | 32.0 | 46 | 92.0 | |
| Incorrect/Incomplete answer | 4 | 8.0 | 4 | 8.0 | |
| Wrong answer | 5 | 10.0 | 0 | 0.0 | |
| Don't know | 25 | 50.0 | 0.0 | 0.0 | |
| **Personal need for training on professional nursing shared governance** | | | | | McNp = 1.000 |
| Yes | 43 | **86.0** | 43 | 86.0 | |
| No | 7 | 14.0 | 7 | 14.0 | |
| **Perception of relation between professional nursing shared governance and nursing practices excellence.** | | | | | – |
| Yes | 50 | 100.0 | 50 | 100.0 | |
| No | 0 | 0.0 | 0 | 0.0 | |

MH, marginal homogeneity test; McN, McNemar test; *, statistically significant at p ≤ 0.05

**Table 4. Effect of awareness sessions on first-line nurse managers' knowledge of professional nursing shared governance concept.**

| Studied variable | Mean ± SD | | |
| --- | --- | --- | --- |
| | Before | After | t p |
| Overall knowledge of professional nursing shared governance | 3.12 ± 2.54 | 5.86 ± 0.35 | 7.493 (<0.001*) |
| Mean % | 52% | 97.67% | |

t, Paired t-test; *, statistically significant at p ≤ 0.05.

**Table 5. Evaluation of First-line nurse managers' satisfaction about professional nursing shared governance awareness session (N = 50).**

| Items of Evaluation | Overall (n = 50) |
| --- | --- |
| | Mean ± SD |
| 1. The session clarified my potential career pathway and future role | 3.82 ± 0.73 |
| 2. The session has increased my strengths and self-awareness. | 4.15 ± 0.77 |
| 3. I am confident that I am obtaining the required knowledge from this session to improve my career | 3.70 ± 1.03 |
| 4. The session increased my feeling of my leadership role toward my staff | 4.41 ± 0.49 |
| 5. The teaching methods used in this session were helpful and effective | 4.02 ± 0.73 |
| 6. The teaching materials and resources used in this session were motivating and helped me to learn. | 3.63 ± 0.98 |
| 7. I enjoyed how my instructor taught the session. | 3.83 ± 0.93 |
| 8. The way my instructor taught the session was suitable to the way I learn. | 3.93 ± 0.79 |
| 9. I hope to repeat this session to all nurses to prepare them for professional nursing shared governance practices as the concept can be applied in practice. | 4.02 ± 0.87 |
| 10. Generally, I am satisfied with module teaching and I hope its recurrence | 3.93 ± 0.79 |
| Overall evaluation of professional nursing shared governance awareness session | |
| Min. – Max. | 3.10 – 4.90 |
| Mean ± SD. | **3.94 ± 0.67** |

## 4. Discussion

Advanced nursing management abilities and competencies are needed to oversee and manage a healthcare organization in a complicated healthcare setting [25–26]. The absence of an effective leadership channel has been identified as a major concern in nursing today, despite the fact that the necessity to practice participatory decision making among different levels of nurses using the shared governance technique has been acknowledged [27–29]. The researchers detected a need for nursing shared governance training and practice because there had not previously been a nursing governance project to develop nursing leaders at the study setting. In order to determine whether a Professional Nursing Shared Governance Awareness Session, as a shared governance initiative, promotes the First-Line Nurse Managers' awareness of the Professional Nursing Shared Governance Concept, the current research was assumed to investigate this.

The current research found that participation in and attendance at awareness sessions significantly increased first-line nurse managers' knowledge of nursing shared governance. Responses indicated that the awareness sessions serve as a basis for this awareness while also advancing knowledge of the nursing shared governance idea. Significantly stating that they require a defined program for professional nursing shared governance were first-line nurse

managers. The majority of first-line nurse managers thought these sessions were excellent and wanted them to be repeated. The majority of first-line nurse managers expressed satisfaction with the knowledge and experience they had gained regarding the nurse manager role, acknowledged that the sessions had improved their strengths and self-awareness, gave their leadership role more importance, and suggested that these sessions be repeated with all nurses to better prepare them for professional and leadership roles. Levels of self-confidence and comprehension of their managing and leadership responsibilities may have been impacted through learning, acquiring knowledge, sharing the experience, and spending time with the various nurse managers. These results were consistent with Gil-Lacruz et al.(2019) [30] who revealed that the health care management related training programs had a useful impact on improvement of the quality of health care as well as organizational development.

Furthermore, Hijazi(2020) [31] conducted a study about governance principles and revealed that there was a statistically significant effect of applying good governance principles on job satisfaction among employees in the public sector.

The difference between the pre- and post-knowledge questionnaire evaluations, according to the study's hypothesis, is evidence of learning that was put into practice and supported H1. Which is also a reflection of the original PICO question: Does the Alexandria Medical Research Institute hospital's first-line nurse managers' knowledge of the professional nursing shared governance concept increase as a result of the awareness sessions? Based on this feedback, the project's awareness workshops might be viewed as a pilot study that will help the hospitals develop a structured, long-term shared governance program.

The findings from this study support those from earlier publications. According to Rundquist (2014) [32], taking part in the professional nursing shared governance awareness training gave participants additional benefits. All study participants agreed that the content presented was important to their learning needs and that the training methods and presentations were successful. Participants in Choi's study (2015) [33] also praised the program's inspirational perspective, usefulness in building a network of resources, facilitation of real-world examples, respectability in building relationships, and facilitation of knowledge dissemination that would advance management understanding and improve leadership role transparency.

These results support what has been suggested in the literature about the value of professional shared governance for nurse managers. In a similar manner, studies conducted by Brahmana, Brahmana, & Ho (2018) [34], Lu, P., Cai, Wei, Song, & Wu (2019) [35], and Mahmood et al. (2018) [36] found that participation in the shared governance program resulted in statistically significant improvements in study subjects' perceptions of their managerial skill, roles, and leadership behaviours. Furthermore, Marais and Petersen (2015) and Sherb et al.(2011) [37,38] demonstrated that a professional nursing shared governance with a career plan could enable work performances that are goal-oriented and focused on improvement in the management skills including communication, delegation, transparency, partnership, equity, accountability, and ownership, which could improve the standard of patient care and the excellence of nursing practices.

Many participants in the previous nursing shared governance awareness sessions ignored a crucial point and recommendation that participatory decision-making and shared governance culture should be communicated through effective training on applying the appropriate model of shared governance that fit their organizational mission, vision, and goals. This training should not be limited to managerial positions but should instead include all nurses as it gives the organization a chance to facilitate. Nurse managers, according to Scherb, Specht, Loes, & Reed (2011) [36], signalled that the approach of shared governance was only applicable to high positions (executive) and did not apply to front and middle nursing management.

In a prior study, Moore, & Wells (2010) [39] conducted a nursing shared governance program for nurses who reported that the program was beneficial, improved their sense of career planning, contributed to their improved understanding of their career pathway (clinical or managerial), a learning prospect to perceive more about the organization, and the specific roles in the hospital. In accordance, Kanninen et al.(2019) [40] revealed that applying nursing shared governance structures into an organization improves the professional practice environment of nursing personnel. In addition, Reddy et al.(2020) [41] concluded that applying an artificial intelligent governance model in health care will lead to good managing structure and improved quality of care and it will affect the improvement of the healthcare providers satisfaction.

In this regard, LaCross et al.(2019), Alyami et al.(2019), ElDemerdash (2024), and DeVellis (2017) [42–45] discussed how developing the next generation of managers and nurses is a comprehensive approach for using participatory decision-making and other shared governance models. To provide high-quality treatment, healthcare institutions must encourage a new generation of nurses to be willing to join and contribute in decision-making. To improve labour force that can meet existing and future healthcare needs, nurse administrators' support is crucial.

The future of the nursing profession depends on shared governance planning and participatory decision-making, which should be a part of every organization's strategic plan. Nursing must continue to integrate the acknowledged benefits of shared governance into evidence-based initiatives and practices. The awareness sessions described in this study might be seen as a pilot study that will help with both current studies detailing the effects of shared governance programs and future studies for programs that are already planned.

According to the study's conclusions, more research is required to better understand the planning, preparation, and dissemination of best practices in nurse shared governance models. In order to strengthen and build official nursing shared governance techniques and models that could increase the retention of the various levels of capable, proficient, and skilled nurses, we advise employing Banner's Novice to Expert model of competency. A formal shared governance model will also appeal to and motivate novices as they grow to capable, proficient, and skilled levels. The implementation of the shared governance program, participant improvement, patient care quality, nursing excellence, and the perception of the nursing profession can all be addressed in follow-up sessions. In order to continue raising awareness of this issue and ensuring the establishment of effective nursing shared governance, researchers, educators, and nursing executives must be given opportunities to present nursing shared governance strategies at numerous meetings, including academic meetings.

Despite the positive outcomes, there are several restrictions that must be acknowledged when doing such research. Only first-line nurse managers were included in the carefully chosen sample size for this study, so it may not be possible to generalize the findings to other industries. However, the results could be improved by holding nursing shared governance awareness sessions for nurses at various levels and in a variety of healthcare settings. The study was only able to look at one component, but future studies could expand on it by tying it to other variables including leadership styles, organizational resilience, and transformation readiness. The implementation of the sessions was also constrained by time. Increasing the sample size, encompassing different sites, and using a control group for comparison could strengthen upcoming studies using the appropriate nurse shared governance model. Alternative research techniques, such as longitudinal case studies and mixed-methods studies, may provide in-depth and more summary information regarding the anticipated relationships with various management principles.

## Conclusion

The goal of this evidence-based practice study was to ascertain the effect of Professional Nursing Shared Governance Awareness Sessions on First-Line Nurse Managers' Knowledge of the Professional Nursing Shared Governance Concept, which were conducted from June to August, 2022. Although the sample size for this study was modest (50 nurse managers), pre- and post-test results showed positive change, showing that professional nursing shared governance awareness training can improve knowledge of shared governance. This study emphasizes the value of continuing to develop knowledgeable nursing leaders. The results supported both *H1* and *PICO* by confirming the significant improvement in the first-line nurse managers' shared governance knowledge after attending the awareness sessions. Participants were satisfied with the intervention and recommend it for all nurses.

## Limitation of the study

The study's sample was restricted to a single study setting and a single pre-and post-test, which limits how broadly the findings may be applied. Furthermore, the current results could contain subjectivity and response bias because they are based on self-reported data. Considering this limitation, the study's conclusions have practical ramifications for how online training and learning programs are designed, particularly with regard to assessment procedures. These constraints might be addressed by longer-term, multi-site experiments that include intervention and control groups and analyse the dynamics of clinical services to assess the efficacy and efficiency of the procedures.

## Author contributions

**Conceptualization:** Sally Mohammed Farghaly Abdelaliem.

**Data curation:** Khalid Al-Mugheed.

**Formal analysis:** Sally Mohammed Farghaly Abdelaliem, Majdi M. Alzoubi, Islam Al-Oweidat, Ghada Mohamed Hamouda.

**Investigation:** Khalid Al-Mugheed, Majdi M. Alzoubi.

**Methodology:** Sally Mohammed Farghaly Abdelaliem.

**Project administration:** Sally Mohammed Farghaly Abdelaliem.

**Visualization:** Khalid Al-Mugheed.

**Writing – original draft:** Sally Mohammed Farghaly Abdelaliem.

**Writing – review & editing:** Sally Mohammed Farghaly Abdelaliem, Islam Al-Oweidat, Ghada Mohamed Hamouda.

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
