## [Decision Letter · Decision Letter 0]

4 Sep 2023

PONE-D-23-02408The Impact of Nursing Shared Governance Educational Program on Women Nurse Managers’ Knowledge for Sustainable Nursing Excellence and Empowerment.PLOS ONE

Dear Dr. Dr. Sally Mohammed Farghaly Abdelaliem,

Thank you for submitting your manuscript to PLOS ONE. After careful consideration, we feel that it has merit but does not fully meet PLOS ONE’s publication criteria as it currently stands. Therefore, we invite you to submit a revised version of the manuscript that addresses the points raised during the review process. Please submit your revised manuscript by Oct 19 2023 11:59PM. If you will need more time than this to complete your revisions, please reply to this message or contact the journal office at plosone@plos.org . Please include the following items when submitting your revised manuscript:

We look forward to receiving your revised manuscript.

Kind regards,

Omar M Khraisat, Associate Professor

Academic Editor

PLOS ONE

Journal Requirements:

"yes 

The research was funded by Princess Nourah bint Abdulrahman University Researchers Supporting Project number (PNURSP2023R279), Princess Nourah bint Abdulrahman University, Riyadh, Saudi Arabia."

Reviewers' comments:

Reviewer's Responses to Questions

**Comments to the Author**

1. Is the manuscript technically sound, and do the data support the conclusions?

Reviewer #1: Partly

Reviewer #2: Yes

2. Has the statistical analysis been performed appropriately and rigorously? 

Reviewer #1: Yes

Reviewer #2: Yes

3. Have the authors made all data underlying the findings in their manuscript fully available?

Reviewer #1: Yes

Reviewer #2: No

4. Is the manuscript presented in an intelligible fashion and written in standard English?

Reviewer #1: No

Reviewer #2: No

5. Review Comments to the Author

Reviewer #1: Overall, I believe that the topic of shared governance is relevant to nurses and administrators around the world and that there are still many steps to take everywhere. Compliments therefore to the authors for taking up this topic in their environment. However, as to whether the main claim of this manuscript offers a significant contribution to the field of nursing management and organization of care, I have some reservations. My main reservation is that the conclusions of this study are rather logical which makes me doubt the scientific relevance. The goal of the intervention was increased awareness around shared governance and this was indeed achieved, measured immediately after the last meeting. Choosing awareness as an outcome variable of an educational intervention does not add much to the scientific discussions around nurse shared governance, after all, increases in knowledge and awareness about a topic immediately after an intervention is logical. An addition of longitudinal measurements (does awareness remain high over time) or change in behavior in practice (e.g. management style) would have added more value to the scientific discourse. The authors acknowledge this limitation already in the discussion section.

- The title mentions that it is about female managers, however, the relevance of this explicit addition is not made clear in the manuscript. I recommend adjusting the title.

- The reference style does not conform to PLOSONE's guidelines. Recommend to use a reference program.

- A reporting checklist or guideline for conducting study and/or paper is missing.

- The introduction is long and not always succinctly described. For example, the first three sentences are wordy. It is not until sentence 4 that it becomes clear what this paper could be about. Advice is to critically review the introduction and write more concisely.

- In lines 280-290 is referred to a previous study of the authors, however the reference is unclear. Precisely this information is interesting because after the introduction on shared governance, I as a reader am also particularly keen to read about the state of shared governance and nursing management and organizational structure in the research setting/country. This knowledge helps a reader to assess whether I could generalize the results of this study to my setting. Nor is this explained anywhere else in the paper.

- The authors use self-designed questionnaires. About the 'professional nursing shared governance knowledge questionnaire', it is written that it was designed based on literature. There is no reference to which literature. For the 'evaluation' questionnaire it is not described on which basis it was designed. Both questionnaires are missing, no reference also to a supplementary file. With regard to validity, it was stated that face and content validity had been established, but information on which technique was used for this was lacking. Nothing was said about internal consistency or structural validity. This limits the opportunities for replication of the studies methodology for other researchers.

- The tables are clear in structure and design

- The discussion is described in a lengthy manner. I advise the authors to improve readability by making the text more concise.

- The paper is about nurse managers, however in the discussion literature of both nurses and nurse managers is used interchangeably. The rationale for this is sometimes lacking. For example in line 636-641.

Based on the intervention, outcome variable and study design, reflection on effect of training on the outcome variable 'awareness' would be of added value.

Reviewer #2: I would like to thank the authors for the opportunity to review this manuscript. The subject of the article is of interest to the community specialized in health and nursing. However, there are serious problems with formatting and compliance with the journal's standards (tables, citations, data availability) that have not been met. A revision in English is needed, perhaps with the help of a native speaker.

I am putting forward my suggestions point by point so that a version of greater quality and robustness can be presented.

Title: Could be adjusted to make it more objective and focused on the object of study. As it is not a gender study, why do the authors use the word "women"? There is no emphasis throughout the text.

Introduction:

I missed more current references on the subject, the last ones are from 2019. I suggest that you also add other existing interventions in the literature that have increased knowledge on the subject.

It is necessary to review all citations according to the journal's guidelines. The model used is vancouver numeric.

Line 289 - The citation of the study is not clear, there are only a few names (Farghaly, S. M., El-Bialy, G. G., & Dowidar).

Lines 294 - 297. The use of the PICO strategy is interesting, but in this case there would be no comparator, as no analysis is carried out with professionals who are not part of the session. I suggest removing or suppressing the use of C.

Method:

Lines 313-316: Add more information about the institution, for example: number of beds, professionals and level of complexity.

Lines 321-324: Were all the nurse managers invited and did they agree to take part in the survey? Do you work with the population?

Lines 328-354: It would be important for the instruments as well as the database to be added as supplementary documents for a better understanding of how the information was collected.

Lines 356-361: What were the Cronbach's alpha values? I suggest presenting them. And better detail the cut-off value for deciding consensus between the judges and how many rounds of consensus were necessary. How were these experts selected? Describe this section better.

Line 363: Change the title to description of the intervention.

Table 1: Could contain more details of the intervention. For example: types of educational techniques, how many meetings, how long the sessions lasted and the number of participants in each (It's not clear whether all 50 managers took part in all the sessions).

Results:

Line 460: Exclude "predominantly female" wouldn't that be all female? after all there were no male participants in the study.

The results could be written without dividing them into sections, so they don't flow and break the reader's train of thought.

Table 3 - I don't understand why the previous information question is important, since after the intervention it is logical for everyone to answer yes. In the variable "Sources of information" in the item "other sources", put in the table caption what the other sources are. Also in table 3, the satisfaction results are presented here and in table 5.

It's not clear from the method, for example, because the instrument is not attached, how the complete response of concepts was assessed.

Discussion:

The discussion needs more up-to-date references, as well as reflection on the authors' own studies already carried out at the same institution on the subject (https://www.sciedupress.com/journal/index.php/jnep/article/view/12157/7665, https://www.cureus.com/articles/24566-perception-of-shared-governance-among-registered-nurses-in-ambulatory-care-center-at-a-tertiary-care-hospital-in-saudi-arabia#!/).

The discussion needs more in-depth reflections on the results found, as well as the authors' statements. I think that the limitations and implications for practice sections can be part of the discussion, and they are naturally arrived at without having to be divided up.

Conclusion:

Needs to be more concise and more focused on answering the objective.

6. PLOS authors have the option to publish the peer review history of their article (what does this mean? ). If published, this will include your full peer review and any attached files.

**Do you want your identity to be public for this peer review?** For information about this choice, including consent withdrawal, please see our Privacy Policy .

Reviewer #1: **Yes: ** Susanne M. Maassen

Reviewer #2: No

---

## [Author Response · Author response to Decision Letter 1]

4 Nov 2023

5. Review Comments to the Author

Reviewer #1: Overall, I believe that the topic of shared governance is relevant to nurses and administrators around the world and that there are still many steps to take everywhere. Compliments therefore to the authors for taking up this topic in their environment. However, as to whether the main claim of this manuscript offers a significant contribution to the field of nursing management and organization of care, I have some reservations. My main reservation is that the conclusions of this study are rather logical which makes me doubt the scientific relevance. The goal of the intervention was increased awareness around shared governance and this was indeed achieved, measured immediately after the last meeting. Choosing awareness as an outcome variable of an educational intervention does not add much to the scientific discussions around nurse shared governance, after all, increases in knowledge and awareness about a topic immediately after an intervention is logical. An addition of longitudinal measurements (does awareness remain high over time) or change in behavior in practice (e.g. management style) would have added more value to the scientific discourse. The authors acknowledge this limitation already in the discussion section.

The title mentions that it is about female managers, however, the relevance of this explicit addition is not made clear in the manuscript. I recommend adjusting the title.

Author responses:

It was adjusted as follows: The Impact of Nursing Shared Governance Educational Program on Nurse Managers’ Knowledge for Sustainable Nursing Excellence and Empowerment.

- The reference style does not conform to PLOSONE's guidelines. Recommend to use a reference program.

Author responses:

The introduction was reviewed critically and the required modifications were done.

- A reporting checklist or guideline for conducting study and/or paper is missing.

Author responses:

Trend checklist was utilized and attached

- The introduction is long and not always succinctly described. For example, the first three sentences are wordy. It is not until sentence 4 that it becomes clear what this paper could be about. Advice is to critically review the introduction and write more concisely.

Author responses:

The introduction was reviewed critically and the required modifications were done.

- In lines 280-290 is referred to a previous study of the authors, however the reference is unclear. Precisely this information is interesting because after the introduction on shared governance, I as a reader am also particularly keen to read about the state of shared governance and nursing management and organizational structure in the research setting/country. This knowledge helps a reader to assess whether I could generalize the results of this study to my setting. Nor is this explained anywhere else in the paper.

Author responses:

The reference was clarified as per the valuable recommendations of the respected reviewer.

- The authors use self-designed questionnaires. About the 'professional nursing shared governance knowledge questionnaire', it is written that it was designed based on literature. There is no reference to which literature. For the 'evaluation' questionnaire it is not described on which basis it was designed. Both questionnaires are missing, no reference also to a supplementary file. With regard to validity, it was stated that face and content validity had been established, but information on which technique was used for this was lacking. Nothing was said about internal consistency or structural validity. This limits the opportunities for replication of the studies methodology for other researchers.

Author responses:

Thank you for the valuable and constructive feedback. We confirm that all the required modifications were done as per your valuable guidance and directions.

- The tables are clear in structure and design- The discussion is described in a lengthy manner. I advise the authors to improve readability by making the text more concise.

Author responses:

Thank you for the valuable and constructive feedback. We confirm that all the required modifications were done as per your valuable guidance and directions.

- The paper is about nurse managers, however in the discussion literature of both nurses and nurse managers is used interchangeably. The rationale for this is sometimes lacking. For example in line 636-641.Based on the intervention, outcome variable and study design, reflection on effect of training on the outcome variable 'awareness' would be of added value.

Author responses:

Thank you for the valuable and constructive feedback. We need to clarify that the reason for using nurses and nurse managers in the discussion literature is returning to the nature of the supporting studies that we considered to support our study main findings.

Reviewer #2: I would like to thank the authors for the opportunity to review this manuscript. The subject of the article is of interest to the community specialized in health and nursing. However, there are serious problems with formatting and compliance with the journal's standards (tables, citations, data availability) that have not been met. A revision in English is needed, perhaps with the help of a native speaker.

I am putting forward my suggestions point by point so that a version of greater quality and robustness can be presented.

Authors responses

Dear Respected Professor Reviewer,

Kindly we would like to thank you for the constructive and valuable review and we confirm that we followed all your valuable instructions and fixed it as per your valuable guidance.

Title: Could be adjusted to make it more objective and focused on the object of study. As it is not a gender study, why do the authors use the word "women"? There is no emphasis throughout the text.

Author responses:

It was adjusted as follows: The Impact of Nursing Shared Governance Educational Program on Nurse Managers’ Knowledge for Sustainable Nursing Excellence and Empowerment.

Introduction:I missed more current references on the subject, the last ones are from 2019. I suggest that you also add other existing interventions in the literature that have increased knowledge on the subject.It is necessary to review all citations according to the journal's guidelines. The model used is vancouver numeric.Line 289

Author responses:

Thank you for the valuable and constructive feedback. We confirm that all the required modifications were done as per your valuable guidance and directions.

- The citation of the study is not clear, there are only a few names (Farghaly, S. M., El-Bialy, G. G., & Dowidar).

Author responses:

Thank you for the valuable and constructive feedback. We confirm that all the required modifications were done as per your valuable guidance and directions. All references and its citation were reviewed and modified.

Lines 294 - 297. The use of the PICO strategy is interesting, but in this case there would be no comparator, as no analysis is carried out with professionals who are not part of the session. I suggest removing or suppressing the use of C.

Author responses:

Thank you for the valuable and constructive feedback. We confirm that all the required modifications were done as per your valuable guidance and directions.

Method:

Lines 313-316: Add more information about the institution, for example: number of beds, professionals and level of complexity.

Author responses:

Thank you for the valuable and constructive feedback. We confirm that all the required modifications were done as per your valuable guidance and directions.

Lines 321-324: Were all the nurse managers invited and did they agree to take part in the survey? Do you work with the population?

Author responses:

Thank you for the valuable and constructive feedback. It was explained as follows; The study's participants were chosen using a whole population sampling strategy (purposive sampling technique), and they included all first-line nurse managers who were employed by the hospital and available during the data collection period (N=50 out of 54), with the aim of assessing their understanding of the concept of professional nursing shared governance and their readiness to take part in the study. All first-line nurse managers working in the study setting met the inclusion criteria, while any staff nurse without a management nursing position did not.

Lines 328-354: It would be important for the instruments as well as the database to be added as supplementary documents for a better understanding of how the information was collected.

Author responses:

Thank you for the valuable and constructive feedback. We confirm that all the required modifications were done as per your valuable guidance and directions.

Lines 356-361: What were the Cronbach's alpha values? I suggest presenting them. And better detail the cut-off value for deciding consensus between the judges and how many rounds of consensus were necessary. How were these experts selected? Describe this section better.

Author responses:

Thank you for the valuable and constructive feedback. We confirm that all the required modifications were done as per your valuable guidance and directions.

Line 363: Change the title to description of the intervention.Table 1: Could contain more details of the intervention. For example: types of educational techniques, how many meetings, how long the sessions lasted and the number of participants in each (It's not clear whether all 50 managers took part in all the sessions).

Author responses:

Thank you for the valuable and constructive feedback. We confirm that all the required modifications were done as per your valuable guidance and directions. The required details about the intervention were added as per your valuable guidance and directions

Results:

Line 460: Exclude "predominantly female" wouldn't that be all female? after all there were no male participants in the study.The results could be written without dividing them into sections, so they don't flow and break the reader's train of thought.

Author responses:

Thank you for the valuable and constructive feedback. We confirm that all the required modifications were done as per your valuable guidance and directions.

Table 3 - I don't understand why the previous information question is important, since after the intervention it is logical for everyone to answer yes. In the variable "Sources of information" in the item "other sources", put in the table caption what the other sources are. Also in table 3, the satisfaction results are presented here and in table 5.It's not clear from the method, for example, because the instrument is not attached, how the complete response of concepts was assessed.

Author responses:

Thank you for the valuable and constructive feedback. We confirm that all the required modifications were done as per your valuable guidance and directions. We need to clarify that the question of the previous information was added for the reason of assessing their initial level of knowledge to assess the gap. In addition the overall satisfaction was removed from table 3 as it is presented in details and the required study tools are attached as per your valuable request.

Discussion:

The discussion needs more up-to-date references, as well as reflection on the authors' own studies already carried out at the same institution on the subject (https://www.sciedupress.com/journal/index.php/jnep/article/view/12157/7665, https://www.cureus.com/articles/24566-perception-of-shared-governance-among-registered-nurses-in-ambulatory-care-center-at-a-tertiary-care-hospital-in-saudi-arabia#!/).The discussion needs more in-depth reflections on the results found, as well as the authors' statements. I think that the limitations and implications for practice sections can be part of the discussion, and they are naturally arrived at without having to be divided up.

Author responses:

Thank you for the valuable and constructive feedback. We confirm that all the required modifications were done as per your valuable guidance and directions.

Conclusion:

Needs to be more concise and more focused on answering the objective.

Author responses:

Thank you for the valuable and constructive feedback. We confirm that all the required modifications were done as per your valuable guidance and directions.

Thank You

---

## [Decision Letter · Decision Letter 1]

28 Feb 2024

PONE-D-23-02408R1The Impact of Nursing Shared Governance Educational Program on Nurse Managers’ Knowledge for Sustainable Nursing Excellence and Empowerment.PLOS ONE

Dear Dr. Farghaly Abdelaliem,

Thank you for submitting your manuscript to PLOS ONE. After careful consideration, we feel that it has merit but does not fully meet PLOS ONE’s publication criteria as it currently stands. Therefore, we invite you to submit a revised version of the manuscript that addresses the points raised during the review process.

We look forward to receiving your revised manuscript.

Kind regards,

Omar Mohammad Ali Khraisat, Associate Professor

Academic Editor

PLOS ONE

Journal Requirements:

Reviewers' comments:

Reviewer's Responses to Questions

**Comments to the Author**

1. If the authors have adequately addressed your comments raised in a previous round of review and you feel that this manuscript is now acceptable for publication, you may indicate that here to bypass the “Comments to the Author” section, enter your conflict of interest statement in the “Confidential to Editor” section, and submit your "Accept" recommendation.

Reviewer #3: (No Response)

Reviewer #4: (No Response)

2. Is the manuscript technically sound, and do the data support the conclusions?

Reviewer #3: Yes

Reviewer #4: Partly

3. Has the statistical analysis been performed appropriately and rigorously? 

Reviewer #3: Yes

Reviewer #4: Yes

4. Have the authors made all data underlying the findings in their manuscript fully available?

Reviewer #3: Yes

Reviewer #4: Yes

5. Is the manuscript presented in an intelligible fashion and written in standard English?

Reviewer #3: Yes

Reviewer #4: Yes

6. Review Comments to the Author

Reviewer #3: Regards

Congratulations to the authors of the article

The necessary reforms that the reviewers had proposed have been applied.

It would have been better to highlight the corrections in the revision of the article.

There is still little information about the validity and reliability of the questionnaire. Nothing was said about internal consistency or structural validity. Describe this section better. Because data collection tool, which is a questionnaire, more explanations are needed.

Thank You

Reviewer #4: Despite the effort to adapt the article based on the recommendations previously issued (strengthening it from a scientific point of view and the originality) and relevance in approaching this subject in a geographical area, which is still little studied, the article maintains its main weakness, since the way in which it is approached does not add to the knowledge that already exists on the subject. Analyzing what someone knows after undergoing a training process and realizing that they know more is the logical outcome for a training process. Perhaps an investigation that reports these changes on the ground, comparing before and after the formative process, would be a great contribution to scientific evolution. In this sense, analyzing the conclusion of the study, without evaluating the impact that the training of professionals had in the field, it is not possible to infer that -"consecutively, result in a positive workplace".

Only evaluating the results in the evolution of dynamics in clinical services can contribute to evaluating the effectiveness and efficiency of these processes.

7. PLOS authors have the option to publish the peer review history of their article (what does this mean? ). If published, this will include your full peer review and any attached files.

**Do you want your identity to be public for this peer review?** For information about this choice, including consent withdrawal, please see our Privacy Policy .

Reviewer #3: No

Reviewer #4: No

---

## [Author Response · Author response to Decision Letter 2]

23 Apr 2024

Reviewer #3: Regards

Congratulations to the authors of the article.

The necessary reforms that the reviewers had proposed have been applied.

It would have been better to highlight the corrections in the revision of the article.

There is still little information about the validity and reliability of the questionnaire. Nothing was said about internal consistency or structural validity. Describe this section better. Because data collection tool, which is a questionnaire, more explanations are needed.

Author Responses:

Dear Respected professor reviewer,

Thank you for the constructive feedback and we confirm that the required modifications were done as following and it was highlighted in yellow within the manuscript:

Tool Adaptation

The study's instruments were created in English and then translated into Arabic using a forward-backward translation technique (Alyami et al. 2019). Five academic experts, whose areas of expertise included nursing administration and health governance, reviewed the tools for translation, face and content validity, and relevance. As a result, the questionnaire underwent little changes. To assess their validity and reliability, we used a variety of techniques. For example, we developed the coding instrument by applying Steps 1–6 of DeVellis' (2017) paradigm for scale construction (refer to Figure 1). Reviewing the literature study (10–12, 20–24) helped to clarify the construct and establish an item pool. Furthermore, Cronbach's alpha and content validity were used.

Validity of content

Five experts from the field, comprising two professors of nursing administration and three professors of health governance. The panel found typographical, punctuation, and word choice issues. In response to their recommendations, a few terms were modified. The knowledge questionnaire's stability over time was investigated in a pilot study with ten nurse supervisors to confirm the instruments' accuracy and functionality and to ascertain the time required to complete the questionnaires. The results revealed a high positive significant correlation (r ranged from 0.782 to 0.890). The pilot sample was absent from the study sample.

Internal consistency

In order to evaluate the instruments' internal consistency, Cronbach's alpha was employed. It is a gauge for how well a collection of items represents a particular concept or dimension. more numbers indicate more reliability and internal consistency. The range is 0 to 1. The professional nursing shared governance knowledge questionnaire had a Cronbach's alpha rating of 0.89. This outcome shows that the study instruments' internal consistency and dependability are satisfactory.

Reviewer #4: Despite the effort to adapt the article based on the recommendations previously issued (strengthening it from a scientific point of view and the originality) and relevance in approaching this subject in a geographical area, which is still little studied, the article maintains its main weakness, since the way in which it is approached does not add to the knowledge that already exists on the subject. Analyzing what someone knows after undergoing a training process and realizing that they know more is the logical outcome for a training process. Perhaps an investigation that reports these changes on the ground, comparing before and after the formative process, would be a great contribution to scientific evolution. In this sense, analyzing the conclusion of the study, without evaluating the impact that the training of professionals had in the field, it is not possible to infer that -"consecutively, result in a positive workplace".

Only evaluating the results in the evolution of dynamics in clinical services can contribute to evaluating the effectiveness and efficiency of these processes.

Author responses:

Dear Respected Professor Reviewer,

Thank you for the valuable and constructive feedback and we confirm that we modified the conclusion section within the abstract as we removed the following sentence: -"consecutively, result in a positive workplace". We modified the title from impact to effect. In addition, we clarified the limitation of our study in a section titled limitation of the study based on your valuable recommendations as following:

The study's sample was restricted to a single study setting and a single pre-and post-test, which limits how broadly the findings may be applied. Furthermore, the current results could contain subjectivity and response bias because they are based on self-reported data. Considering this limitation, the study's conclusions have practical ramifications for how online training and learning programs are designed, particularly with regard to assessment procedures. These constraints might be addressed by longer-term, multi-site experiments that include intervention and control groups and analyse the dynamics of clinical services to assess the efficacy and efficiency of the procedures.

Thank you

---

## [Decision Letter · Decision Letter 2]

11 Jul 2024

The Effect of Nursing Shared Governance Educational Program on Nurse Managers’ Knowledge for Sustainable Nursing Excellence and Empowerment.

PONE-D-23-02408R2

Dear Dr.,

We’re pleased to inform you that your manuscript has been judged scientifically suitable for publication and will be formally accepted for publication once it meets all outstanding technical requirements.

Kind regards,

Omar Mohammad Ali Khraisat, Associate Professor

Academic Editor

PLOS ONE

Additional Editor Comments (optional):

Reviewers' comments:

Reviewer's Responses to Questions

**Comments to the Author**

1. If the authors have adequately addressed your comments raised in a previous round of review and you feel that this manuscript is now acceptable for publication, you may indicate that here to bypass the “Comments to the Author” section, enter your conflict of interest statement in the “Confidential to Editor” section, and submit your "Accept" recommendation.

Reviewer #3: All comments have been addressed

2. Is the manuscript technically sound, and do the data support the conclusions?

Reviewer #3: Yes

3. Has the statistical analysis been performed appropriately and rigorously? 

Reviewer #3: Yes

4. Have the authors made all data underlying the findings in their manuscript fully available?

Reviewer #3: Yes

5. Is the manuscript presented in an intelligible fashion and written in standard English?

Reviewer #3: Yes

6. Review Comments to the Author

Reviewer #3: Regards

Congratulations to the authors of the article. The necessary reforms that the reviewers had proposed have been applied.

Thank You.

7. PLOS authors have the option to publish the peer review history of their article (what does this mean? ). If published, this will include your full peer review and any attached files.

**Do you want your identity to be public for this peer review?** For information about this choice, including consent withdrawal, please see our Privacy Policy .

Reviewer #3: No

---

## [Editor Report · Acceptance letter]

PONE-D-23-02408R2

PLOS ONE

Dear Dr. Farghaly Abdelaliem,

I'm pleased to inform you that your manuscript has been deemed suitable for publication in PLOS ONE. Congratulations! Your manuscript is now being handed over to our production team.

Kind regards,

on behalf of

Dr. Omar Mohammad Ali Khraisat

Academic Editor

PLOS ONE